# Collective total synthesis of C4-oxygenated securinine-type alkaloids via stereo-controlled diversifications on the piperidine core

Sangbin Park [1], Gyumin Kang[1,2], Chansu Kim[1], Dongwook Kim [2] & Sunkyu Han [1,2] ✉

Securinega alkaloids have fascinated the synthetic chemical community for over six decades. Historically, major research foci in securinega alkaloid synthesis have been on the efficient construction of the fused tetracyclic framework that bears a butenolide moiety and tertiary amine-based heterocycles. These "basic" securinega alkaloids have evolved to undergo biosynthetic oxidative diversifications, especially on the piperidine core. However, a general synthetic solution to access these high-oxidation state securinega alkaloids is lacking. In this study, we have completed the total synthesis of various C4-oxygenated securinine-type alkaloids including securingines A, C, D, securitinine, secu'amamine D, phyllanthine, and 4-epi-phyllanthine. Our synthetic strategy features stereocontrolled oxidation, rearrangement, and epimerization at N1 and C2–C4 positions of the piperidine core within (neo)securinane scaffolds. Our discoveries provide a fundamental synthetic solution to all known securinine-type natural products with various oxidative and stereochemical variations around the central piperidine ring.

Securinega alkaloids have fascinated chemists for over 60 years since the first isolation of securinine[1,2]. Plants that contain these alkaloids have been applied in traditional medicines for the treatment of malaria and other diseases[3–5]. The basic securinine framework can undergo biosynthetic oxidations to yield various natural products with oxidative decorations around the piperidine moiety. In that regard, recent isolation campaigns from *Securinega suffruticosa* resulted in an outburst of discoveries of novel C4-oxygenated securinine-type alkaloids that show oxidative and (or) stereochemical variations around the piperidine ring A (Fig. 1)[6–11]. With respect to the structure determination of natural products, the continued development of computational chemistry has enabled the estimation of spectroscopic data of complex molecules[12]. Recently, we proposed an alternative structure of securingine A (**7**) based on its calculated ground-state structure and

DP4 + probability analysis[13]. Kutateladze and coworkers suggested structure revisions of securingines C (**8**) and D (**9**) based on their machine learning (ML)-based hybrid density functional theory/parametric computations of nuclear magnetic resonance spectra[14]. However, these computationally proposed structures have yet to be experimentally confirmed via total synthesis.

The tetracyclic structure of securinega alkaloids that features butenolide and tertiary amine moieties has served as an excellent platform for the development of new synthetic strategies and tactics[15–17]. Over 25 reports have depicted the synthesis of securinine-type alkaloids with piperidine-based A-ring. Among those works, Busqué and de March's total synthesis of allosecurinine (**2**) that featured a vinylogous Mannich reaction[18] between hemiaminal **10** and silyl dienol ether derivative **11** and an intramolecular *N*-alkylation

[1]Department of Chemistry, Korea Advanced Institute of Science & Technology (KAIST), Daejeon 34141, Republic of Korea. [2]Center for Catalytic Hydrocarbon Functionalizations, Institute for Basic Science (IBS), Daejeon 34141, Republic of Korea. ✉e-mail: sunkyu.han@kaist.ac.kr

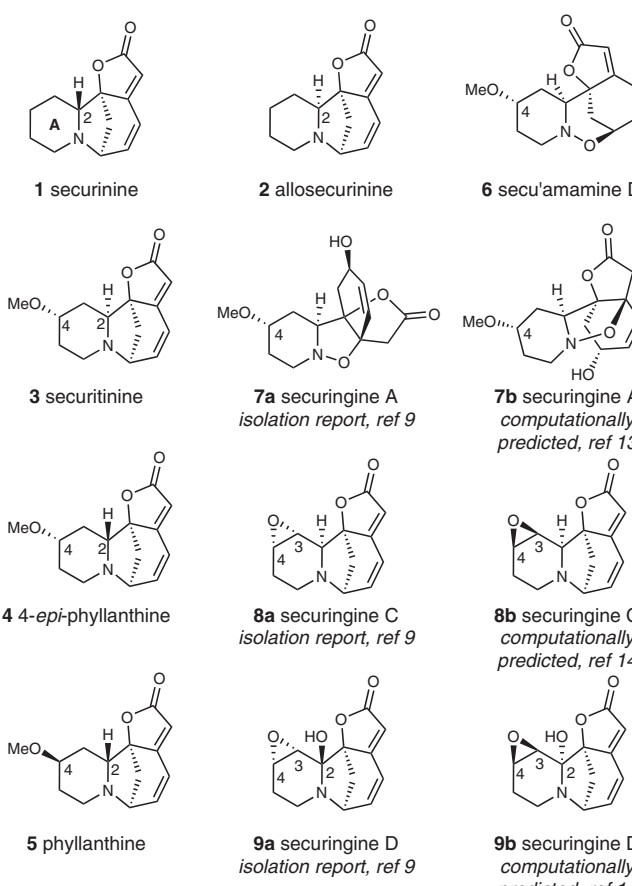

**Fig. 1 | Representative securinine-type natural products of our interest.** Structures of securinine, allosecurinine, and representative C4-oxygenated securinine-type natural products are depicted. Both the originally proposed structure and the computationally predicted structure are presented for securingines A, C, and D.

greatly inspired us at the onset of our synthetic campaign by providing us with guiding footprints on how to efficiently assemble the eastern menisdaurilide-based and the western piperidine-based fragments (Fig. 2a)[19]. Furthermore, Gademann's elegant total synthesis of allosecurinine (**2**), which was enabled by an intramolecular *aza*-Michael addition to access secu'amamine E (**13**) and its subsequent mesylation-based 1,2-amine shift (Fig. 2b)[20,21] has enabled us to explore new chemistries within the neosecurinane framework[22]. For the synthesis of tetrahydro-1,2-oxazine-based securinine-type natural products, Horii, Parello and coworkers reported the biomimetic oxidation of allosecurinine (**2**) to phyllantidine (**14**, Fig. 2c) in 1972[23]. Importantly, the tetrahydro-1,2-oxazine core could also be established by the Kerr group via a Lewis acid-catalyzed three-component homo [3 + 2] dipolar cycloaddition[24] and by the Wood group via an acyloxy nitroso ring expansion strategy[25] en route to their total synthesis of phyllantidine (**14**), respectively. With respect to the C4-oxygenated high-oxidation state securinega alkaloid, the Weinreb group's total synthesis of phyllanthine (**5**) via SmI₂-mediated intramolecular ketonitrile coupling and stereoselective imino Diels−Alder reaction has remained as the only synthesis of securinega alkaloid with a C4-oxygenation[26,27].

Building on these seminal prior synthetic studies, we envisioned establishing a general synthetic solution for various C4-oxygenated securinine-type alkaloids. Structural examination of C4-oxygenated securinine-type alkaloids revealed key challenges en route to our grand goal. First, the isolation of both phyllanthine (**5**) and 4-*epi*-phyllanthine (**4**) with *R*- and *S*-configuration at C4, respectively, hinted at the necessity to develop a synthetic strategy that can introduce the methoxy group at C4 with complete stereocontrol (Fig. 2d). Next,

structural comparison of 4-*epi*-phyllanthine (**4**) and securitinine (**3**) unveiled the importance of stereochemical flexibility and control at C2 methine in our synthetic approach. Then, the isolations of both securingine A (**7b**) and secu'amamine D (**6**) highlighted the importance of control over the atom connectivity after the oxidation of the tertiary amine moiety. After that, the isolation of securingine C (**8b**) necessitated a method to stereoselectively epoxidize the C3−C4 site. Finally, structural comparison of securingines C (**8b**) and D (**9b**) asked for a method that can regioselectively oxygenate the C2 position. Herein, we introduce a unified synthetic route to high-oxidation state securinine-type alkaloids that addresses all these key challenges.

## Results
### Decagram-scale synthesis of *O*-silylated menisdaurilide
Our study commenced with the synthesis of the right-hand fragment of the securinine framework. Despite previous reports on enantioselective total synthesis of menisdaurilide[28–31], a practical decagram-scale synthetic process to enantioenriched menisdaurilide or its derivative had not been described when we started our studies. In 2019, Peixoto and coworkers reported an elegant 5-step synthesis of (±)-*O*-*t*-butyldimethylsilylmenisdaurilide that delivered 2.5 g of material in a single pass. However, enantiomerically enriched menisdaurilide derivative could be obtained only after semi-preparative chiral HPLC separation[31]. Resolution of (±)-*O*-*t*-butyldimethylsilylmenisdaurilide via its derivatization with the enantiomerically enriched carboxylic acid was possible but required multiple flash chromatographic separations[32]. Hence, we first aimed to develop a decagram-scale synthesis of enantiomerically enriched menisdaurilide derivative (Fig. 3). Our synthesis commenced with enantioselective catalytic desymmetrization of **15** previously reported by Snapper, Hoveyda, and coworkers[33,34]. Enantioenriched (95% ee) alcohol **18** was subjected to hydroxyl-directed epoxidation reaction conditions to afford epoxide **19** in 98% yield. Alcohol **19** was then allowed to react with Dess−Martin periodinane and subsequently with silica gel to yield γ-hydroxyenone **20** in 94% yield. TBDPS protection of alcohol **20** (95% yield) followed by selective TBS deprotection in the presence of zinc bromide (83% yield) yielded α-hydroxyketone **21**. Finally, after ester formation between **21** and diethylphosphonoacetic acid (96% yield), the resulting phosphonate **22** was treated with potassium carbonate and 18-crown-6 to afford the HWE reaction product, *O*-silylated menisdaurilide derivative **23** (61% yield, 95% ee). It is important to note that all synthetic steps from precursor **15** were conducted on a >10 g scale which enabled the decagram access to *O*-*t*-butyldiphenylsilylmenisdaurilide (**23**) in a single pass (Fig. 3).

### Total synthesis of securingine A and secu'amamine D
With a robust and scalable synthetic access to **23**, we embarked on a divergent synthesis of N−O bond containing securingine A (**7b**) and secu'amamine D (**6**). Our first challenge was the enantioselective introduction of the methoxy group at the C4 position of lactam **24**. In 2009, the Yun group reported an enantioselective conjugate borylation of cyclic enones and lactones catalyzed by copper−Taniaphos complex[35]. Inspired by this report, we conducted asymmetric conjugate borylation of lactam **24** in the presence of B₂pin₂, CuCl, (*S*)-Taniaphos (**25**), and NaO*t*Bu. The resulting boronic ester was oxidized with sodium perborate to yield alcohol **26** in 54% over 2 steps and was subsequently methylated to yield C4-methoxy adduct **27** (72% yield). To our pleasure, chiral ¹H NMR analysis of **27** that lacks a chromophore using cationic cobalt complex developed by the Kim group[36] revealed that the conjugate borylation occurred in 96% ee (Fig. 4). Lactam **27** was then reduced to hemiaminal **28** and allowed to react with silyl dienol ether **11** accessed from **23** under the vinylogous Mannich reaction conditions developed by Busqué, de March, and coworkers to forge the tricyclic structure[19]. Subsequent TBAF-mediated silyl deprotection of the resulting tricycle afforded alcohol **29** in 61% yield

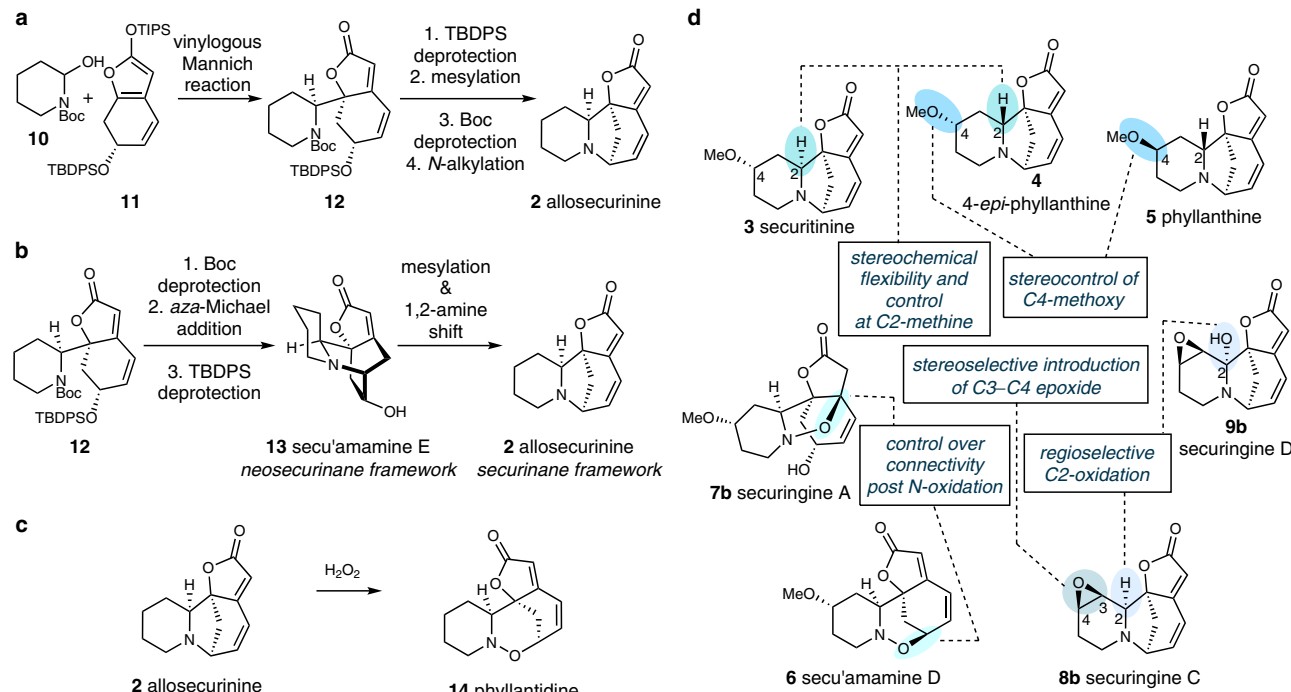

**Fig. 2 | Select prior syntheses and derivatization of allosecurinine and our synthetic blueprint for C4-oxygenated securinine-type natural products.**
**a** Busqué and de March's biosynthetically inspired synthesis of allosecurinine (ref. 19); **b** Gademann's biosynthetically inspired synthesis of secu'amamine E and

allosecurinine (ref. 22); **c** Horii and Parello's biomimetic conversion of allosecurinine to phyllantidine (ref. 23); **d** Pre- and post-modification of the securinega framework (this work). Boc ᵗbutoxycarbonyl, TIPS triisopropylsilyl, TBDPS ᵗbutyldiphenylsilyl.

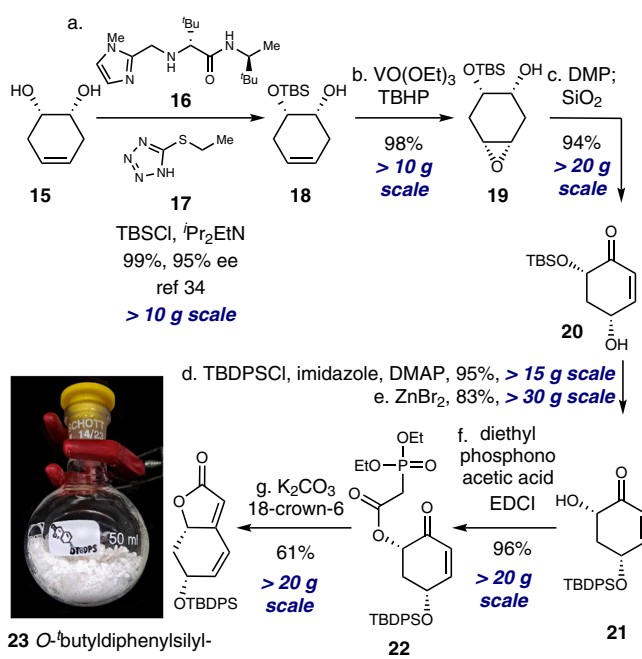

**Fig. 3 | Synthesis of O-tbutyldiphenylsilylmenisdaurilide (23).** Reagents and conditions: (a) **16** (0.2 equiv), **17** (0.1 equiv), TBSCl (2.0 equiv), ⁱPr₂EtN (1.2 equiv), THF, −40 °C, 99% (95% ee); (b) VO(OEt)₃ (0.05 equiv), TBHP (4.0 equiv), CH₂Cl₂, 0 °C to 23 °C, 98%; (c) DMP (1.25 equiv), CH₂Cl₂, 23 °C; SiO₂, 94%; (d) TBDPSCl (1.2 equiv), imidazole (1.2 equiv), DMAP (0.1 equiv), CH₂Cl₂, 0 °C to 23 °C, 95%; (e) ZnBr₂ (5.0 equiv), H₂O (5.0 equiv), CH₂Cl₂, 50 °C, 83%; (f) diethylphosphonoacetic acid (2.0 equiv), EDCI (2.0 equiv), CH₂Cl₂, 23 °C, 96%; (g) K₂CO₃ (5.0 equiv), 18-crown-6 ether (5.0 equiv), THF, 0 °C, 61%. TBS ᵗbutyldimethylsilyl, TBHP ᵗbutyl hydroperoxide, DMP Dess–Martin periodinane, DMAP 4-dimethylaminopyridine, EDCI 1-ethyl-3-(3-dimethylaminopropyl)carbodiimide.

over 2 steps from **23**. Carbamate **29** was subsequently subjected to a reaction sequence that involved TFA-mediated Boc deprotection and base-promoted intramolecular *aza*-Michael reaction to produce tetracycle **30**[22].

To our delight, when amine **30** was treated with *m*-CPBA and potassium carbonate, the first synthetic sample of securingine A (**7b**) was obtained in 85% yield. Spectroscopic data of our synthetic sample of **7b** were in complete agreement with those from the isolation report[9]. Hence, our first total synthesis of securingine A (**7b**) unambiguously corroborated its computationally proposed structural revision[13]. The presumed mechanism of the transformation of **30** to securingine A (**7b**) involves initial *N*-oxidation of **30** to **31** followed by a Cope elimination to result in intermediate **32**. The hydroxyl amine moiety of **32** would then undergo intramolecular 1,4-conjugate addition to give securingine A (**7b**). On the other hand, when securitinine (**3**), accessed by treating **30** with MsCl and Et₃N[20–22,31], was allowed to react with *m*-CPBA and potassium carbonate, secu'amamine D (**6**) was obtained in 75% yield via a 1,2-Meisenheimer rearrangement[23,37]. Mechanistically, it is notable that the formation of relatively more stable allylic radical intermediate enabled homolytic cleavage of C7–N1 in *N*-oxide intermediate **33**[38] (contrary to the heterolytic C−N cleavage of intermediate **31**) and subsequent intramolecular radical recombination to produce secu'amamine D (**6**).

## Total synthesis of securingines C and D

We then turned our attention to the total synthesis of epoxide-containing securinine-type alkaloids securingines C (**8b**) and D (**9b**). After numerous experimentations, we discovered that fructose-derived Shi's ketone **35** could effectively catalyze the epoxidation of α,β-unsaturated lactam **24** in 71% yield (2 cycles) and 3:1 er (Fig. 5)[39]. The resulting epoxide was reduced to hemiaminal intermediate **37** and allowed to react with silyl dienol ether **11** in the presence of ᵗBu₂BOTf to forge the tricyclic intermediate[19], which after TBAF-mediated deprotection of the TBDPS group yielded alcohol **38** (46% isolated yield of

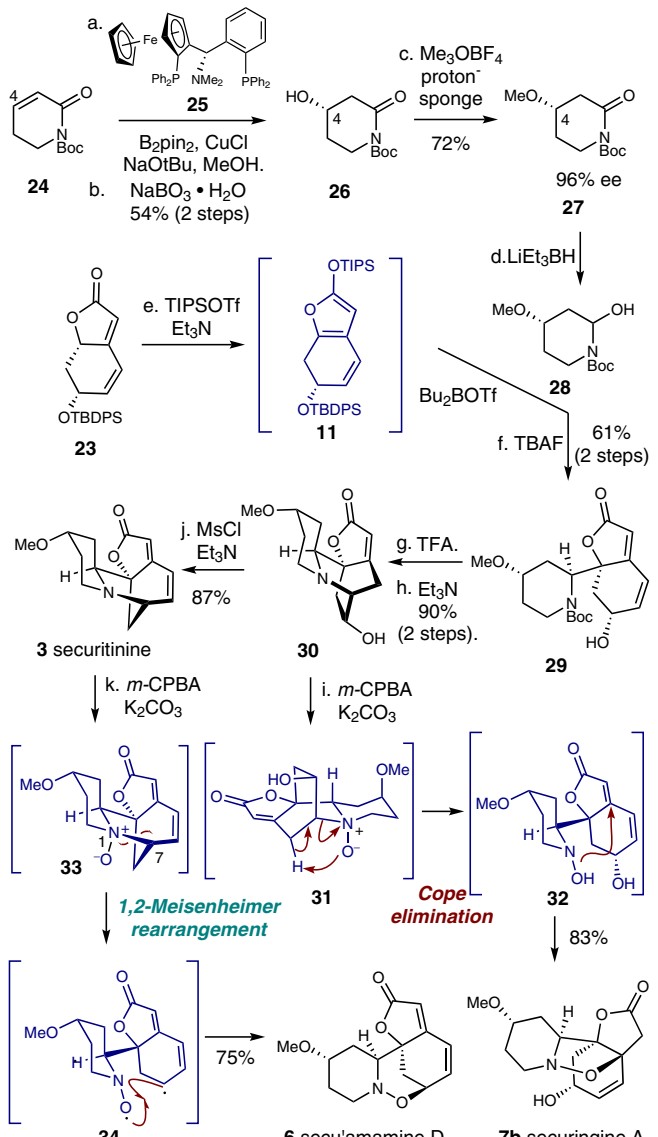

**Fig. 4 | Total synthesis of secu'amamine D (6) and securingine A (7b) via divergent 1,2-Meisenheimer rearrangement and Cope elimination.** Reagents and conditions: (a) B₂pin₂ (1.1 equiv), CuCl (0.02 equiv), NaOᵗBu (0.03 equiv), **25** (0.04 equiv), MeOH (2.0 equiv), THF, 23 °C; (b) NaBO₃·H₂O (5.0 equiv), THF:H₂O (1:1 *v/v*), 23 °C, 54% (2 steps); (c) Me₃OBF₄ (3.0 equiv), proton-sponge (4.0 equiv), CH₂Cl₂, 0 °C to 23 °C, 72% (96% ee); (d) LiEt₃BH (1.2 equiv), THF, −78 °C; (e) TIPSOTf (1.2 equiv), Et₃N (2.0 equiv), Et₂O, 0 °C to 23 °C; **28** (1.5 equiv), Bu₂BOTf (1.2 equiv), Et₂O, −78 °C; (f) TBAF (2.2 equiv), THF, 23 °C, 61% (2 steps); (g) TFA:CH₂Cl₂ (1:1 *v/v*), 23 °C; (h) Et₃N:MeOH (1:2 *v/v*), 50 °C, 90% (2 steps); (i) *m*-CPBA (1.1 equiv), K₂CO₃ (3.0 equiv), CH₂Cl₂, 0 °C to 23 °C, 83%; (j) MsCl (3.0 equiv), Et₃N (6.0 equiv), CH₂Cl₂, 0 °C, 87%; (k) *m*-CPBA (1.1 equiv), K₂CO₃ (3.0 equiv), CH₂Cl₂, 0 °C to 23 °C, 75%. B₂pin₂ bis(pinacolato)diboron, TBAF tetra-*n*-butylammonium fluoride, TFA trifluoroacetic acid, *m*-CPBA 3-chloroperbenzoic acid.

the desired diastereomer). Markedly, the diastereomeric byproduct that originated from the enantiomer of **36** could be readily separated from **38** by silica gel column chromatography which practically solved the issue related with the moderate enantioselectivity during the epoxidation of **24**. The subjection of the resulting alcohol **38** to a three-step sequence involving mesylation of the alcohol moiety, Boc deprotection, and intramolecular *N*-alkylation afforded the first synthetic sample of securingine C (**8b**). Spectroscopic data of the synthetic sample were in complete accordance with the isolation report[9]. Since the relative configuration of the epoxide moiety was unambiguously confirmed by a single-crystal X-ray diffraction analysis of the

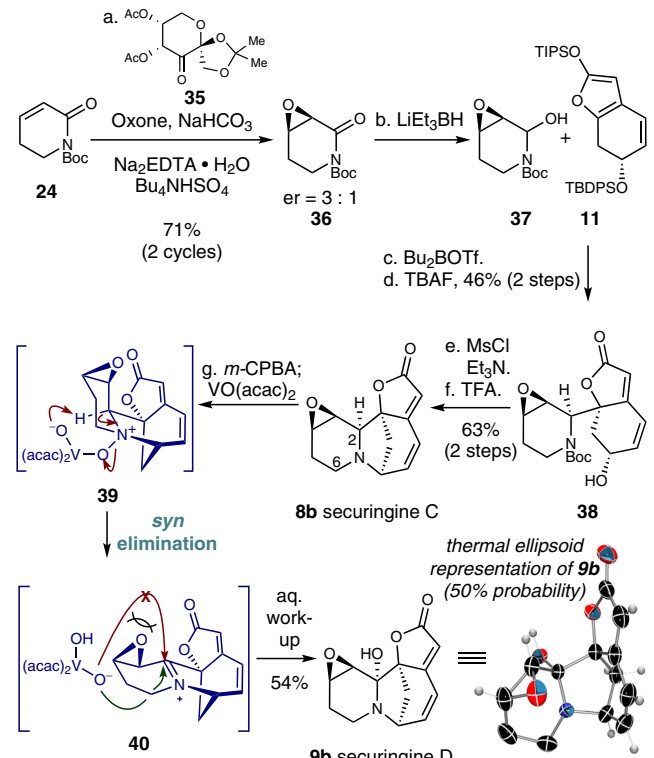

**Fig. 5 | Total synthesis of securingines C (8b) and D (9b).** Reagents and conditions: (a) Oxone (10.0 equiv), NaHCO₃ (30.0 equiv), **35** (0.5 equiv), Na₂EDTA (0.0005 equiv), Bu₄NHSO₄ (0.1 equiv), MeCN, 0 °C to 23 °C, 71% (3:1 er) after 2 cycles; (b) LiEt₃BH (1.2 equiv), THF, −78 °C; (c) Bu₂BOTf (1.2 equiv), Et₂O, −78 °C; (d) TBAF (2.2 equiv), THF, 23 °C, 46% (2 steps); (e) MsCl (3.0 equiv), Et₃N (6.0 equiv), CH₂Cl₂, 0 °C; (f) TFA: CH₂Cl₂ (1:1 *v/v*), 23 °C, 63% (2 steps); (g) *m*-CPBA (1.1 equiv), VO(acac)₂ (1.0 equiv), CH₂Cl₂, 0 °C, 54%. EDTA ethylenediaminetetraacetic acid, VO(acac)₂ vanadyl acetylacetonate.

downstream derivative of **8b** (*vide infra*), we experimentally confirmed the computationally proposed structure revision of securingine C (**8b**) by the Kutateladze group[14].

With efficient access to securingine C (**8b**), its regio- and stereoselective C−H hydroxylation at C2 position was explored. We previously reported a C2-selective enamine formation of (viro)allosecurinine via VO(acac)₂-mediated Polonovski reaction[32]. Based on these results, we treated securingine C (**8b**) with *m*-CPBA and subsequently with VO(acac)₂ to access intermediate **39**. Pleasantly, **39** underwent intramolecular *syn*-elimination to forge C2-iminium ion **40**, the regioisomer that could not be accessed under conventional E2(*anti*-elimination)-based Polonovski reaction conditions[32]. Hydroxyvanadium intermediate consequently trapped iminium ion **40** from the sterically more accessible *re* face to yield the first synthetic sample of securingine D (**9b**) upon aqueous work-up in 54% yield. Spectroscopic data of the synthetic sample were in complete agreement with those of the natural sample[9]. Importantly, single-crystal X-ray diffraction analysis of securingine D (**9b**, CCDC 2170237) unequivocally established the absolute and relative stereochemistry of the synthetic securingine D (**9b**). Hence, the machine learning-based structural revision of securingine D (**9b**)[14] was confirmed via its total synthesis.

## C2-epimerization of securinine-type alkaloids

Next, we sought to find a solution for the C2-epimerization of securinine-type alkaloids. In that regard, we turned our attention to hydrogen atom transfer (HAT)-mediated epimerization reaction[40]. Recently, Ellman, Houk, Mayer, and coworkers reported a combined photocatalytic and HAT-based strategy for the epimerization of C2-substituted piperidine[41]. They showed that the thiophenyl radical

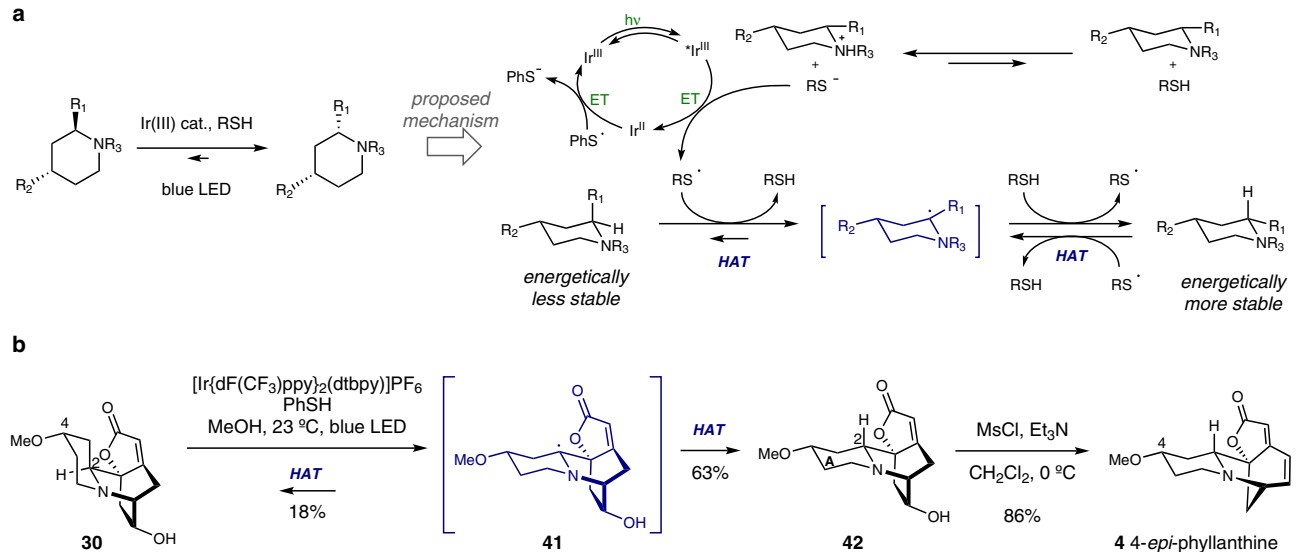

**Fig. 6 | Total synthesis of 4-epi-phyllanthine (4) via hydrogen atom transfer (HAT)-based C2-epimerization of the neosecurinine scaffold. a** Ellman's light-mediated piperidine epimerization (ref. 41); **b** Light-mediated C2-epimerization of **30** enables the synthesis of 4-epi-phyllanthine (**4**). ET electron transfer, HAT hydrogen atom transfer.

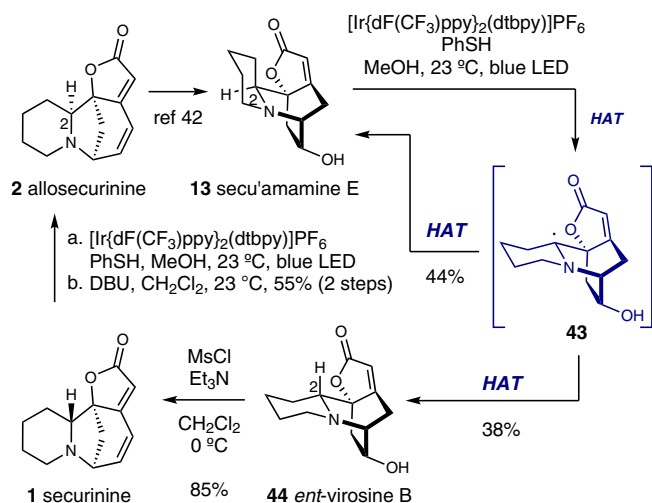

**Fig. 7 | Interconversion between allosecurinine (2) and securinine (1) via HAT-based C2-epimerization.** The light-mediated HAT-based epimerization of piperidines enabled the interconversion between secu'amamine E and ent-virosine B, the neosecurinane derivatives of allosecurinine and securinine, respectively. Direct conversion of securinine to allosecurinine was also possible by the iridium-catalyzed HAT-based epimerization reaction by employing 2 equiv of benzenethiol and subsequent DBU-mediated E1cB reaction. DBU 1,8-diazabicyclo[5.4.0]undec-7-ene.

generated by photoredox catalyst can undergo reversible polarity matched HAT with C2-substituted piperidines to eventually yield the energetically more stable piperidine products (Fig. 6a). Encouraged by this report, we allowed **30** to react with [Ir{dF(CF₃)ppy}₂(dtbpy)]PF₆ (1 mol%) and PhSH (1 equiv) in methanol under blue LED irradiation. To our delight, C2-epimerized product **42** was obtained in 63% yield along with 18% of **30** via radical intermediate **41** (Fig. 6b). DFT-calculated difference of the solution-phase free energy between **42** and **30** supported the favorable formation of **42** over **30**. The calculated ground-state conformation of compound **30** showed that the C4-methoxy moiety resides in the axial position while that of compound **42** is positioned in the equatorial position (See the supporting information

for detail). Indeed, the observed formation ratio between **42** and **30** (**42**:**30** = 3.5:1) was in line with the predicted ratio based on the A-value of the methoxy group (0.6 kcal/mol, corresponding to 2.8:1 ratio at 23 °C). Furthermore, resubjection of **42** to the aforementioned reaction conditions resulted in the formation of **42** and **30** in approximately 3:1 ratio, corroborating the thermodynamic equilibrium between these two species. Subsequent 1,2-amine shift via a mesylation of the resulting hydroxyl moiety within compound **42** afforded the first synthetic sample of 4-epi-phyllanthine (**4**).

The prevalence of C2-epimeric pairs in securinega alkaloids prompted us to test the generality of light-mediated HAT-based epimerization strategy by applying it to (neo)securinane alkaloids without the C4-methoxy moiety. Hence, we investigated the potential interconversion between allosecurinine and securinine. Firstly, allosecurinine (**2**)[32] was transformed to secu'amamine E (**13**) following our previously reported deconstructive functionalization method as a means to mask the Michael acceptor moiety (Fig. 7)[42]. Secu'amamine E (**13**) was then subjected to the photoredox-catalyzed HAT-mediated C2-epimerization reaction conditions and the C2-epimer ent-virosine B (**44**) was obtained in 44% yield along with secu'amamine E (**13**, 38% yield). Importantly, resubjection of ent-virosine B (**44**) to the same reaction conditions resulted in the formation of both **13** and **44** in 38% and 31% yield, consistent with the thermodynamic equilibrium between these two compounds (See the supporting information for DFT-calculation-based analysis). ent-Virosine B (**44**) was then transformed to securinine (**1**) upon mesylation of the alcohol moiety and consequent 1,2-amine shift. Finally, direct C2-epimerization of the securinine framework (azabicyclo[3.2.1]octane core) without the intermediacy of the neosecurinine-type (azabicyclo[2.2.2]octane core) intermediate was explored. Delightfully, when securinine (**1**) was allowed to react with [Ir{dF(CF₃)ppy}₂(dtbpy)]PF₆ (1 mol%) and 2 equiv of benzenethiol, and subsequently with 1,8-diazabicyclo[5.4.0]undec-7-ene (DBU), allosecurinine (**2**) was obtained in 55% yield along with securinine (**1**, 14% yield). The use of 2 equiv of benzenethiol was necessary as one equivalent of it underwent 1,6-conjugate addition to the substrate. DBU was the optimal base for the E1cB elimination of benzenethiol. The prevalence of C2-epimeric pairs within securinega alkaloids family hints at further general transformations between those epimeric pairs.

## Total synthesis of phyllanthine and its C2-epimer

The Mannich reaction-based union of the piperidine precursor (A-ring) and the menisdaurilide derivative **23** has proven to be a robust method to set the *S*-configuration at the C2 position (Figs. 4 and 5)[19]. Peixoto and coworkers could obtain a diastereomeric mixture of compounds with *R*- and *S*-configurations at C2 via their aldol reaction-based fragments coupling and subsequent reductive amination strategy[31]. We also showed the versatility of HAT approach for the epimerization of this stereogenic center (Figs. 6 and 7). However, direct and selective access to the *R*-configuration at C2 via a C2–C9 bond formation between the piperidine precursor and the menisdaurilide derivative **23** has been historically elusive[31]. Hence, we initiated the exploration of an orthogonal C2–C9 bond-forming strategy that selectively sets the *R*-configuration at the C2 position. To our pleasure, we found that the vinylogous Michael addition of the lithium dienolate derivative of **23** to enone **45**, accessed by dehydrogenation of 1-Boc-4-piperidone[43], resulted in the diastereoselective formation of tricycle **46** in 79% yield (Fig. 8a). We reasoned that the sterically bulky TBDPS moiety in the dienolate derivative of **23** governs its facial selectivity and sets the stereochemistry at C9 (Fig. 8b). The steric clash between the C6-methylene of **45** and the menisdaurilide backbone renders the *si*-face approach of the nucleophile disfavorable, hence, leading to the diastereoselective formation of **46** with the *R*-configuration at C2 (Fig. 8b).

With robust access to compound **46**, we envisioned converting it to phyllanthine (**5**). **46** was stereoselectively reduced to alcohol **47** in 84% yield upon reaction with K-selectride. Less sterically hindered sodium borohydride resulted in the same stereochemical outcome. The C4 stereochemistry in **47** was inverted via a two-step Mitsunobu

reaction-mediated protocol to yield alcohol **48** (99% yield over 2 steps). Ensuing methylation of alcohol **48** with Meerwein salt, consequent desilylation, Boc deprotection, and base-mediated intramolecular *aza*-Michael addition of the resulting tricycle **49** afforded tetracyclic compound **50**. Mesylation of the alcohol moiety in **50** induced the 1,2-amine shift to yield phyllanthine (**5**) in 90% yield. As expected, application of previously explored C2-epimerization reaction conditions to **50** resulted in thermodynamically more stable compound **51** in 63% yield along with 15% of **50**. DFT-calculated difference of the solution-phase free energy between **50** and **51** was in line with the favorable formation of **51** over **50**. Final mesylation of alcohol **51** produced 4-*epi*-securitinine (**52**, 72% yield), a presumed natural product yet to be discovered[44].

## Discussion

To sum up, we described a unified synthetic approach to C4-oxygenated securinine-type alkaloids. Securingines A (**7b**), C (**8b**), D (**9b**), securitinine (**3**), secu'amamine D (**6**), 4-*epi*-phyllanthine (**4**), and phyllanthine (**5**) could be chemically synthesized in this study. Keys to our success were; (1) the scalable synthetic route to the menisdaurilide derivative **23**, (2) the discovery of catalytic reaction conditions for asymmetric borylation and epoxidation of α,β-unsaturated lactam, (3) the mechanism-based substrate design of *N*-oxide derivatives favoring either 1,2-Meisenheimer rearrangement or Cope elimination, (4) the VO(acac)₂-mediated Eᵢ (*syn*) selective Polonovski reaction, (5) the application of Ellman's light-mediated HAT-based piperidine epimerization, and (6) the diastereoselective Michael addition of the lithium dienolate derivative of menisdaurilide to piperidone **45**. These findings have built the foundation to access all known C4-oxygenated

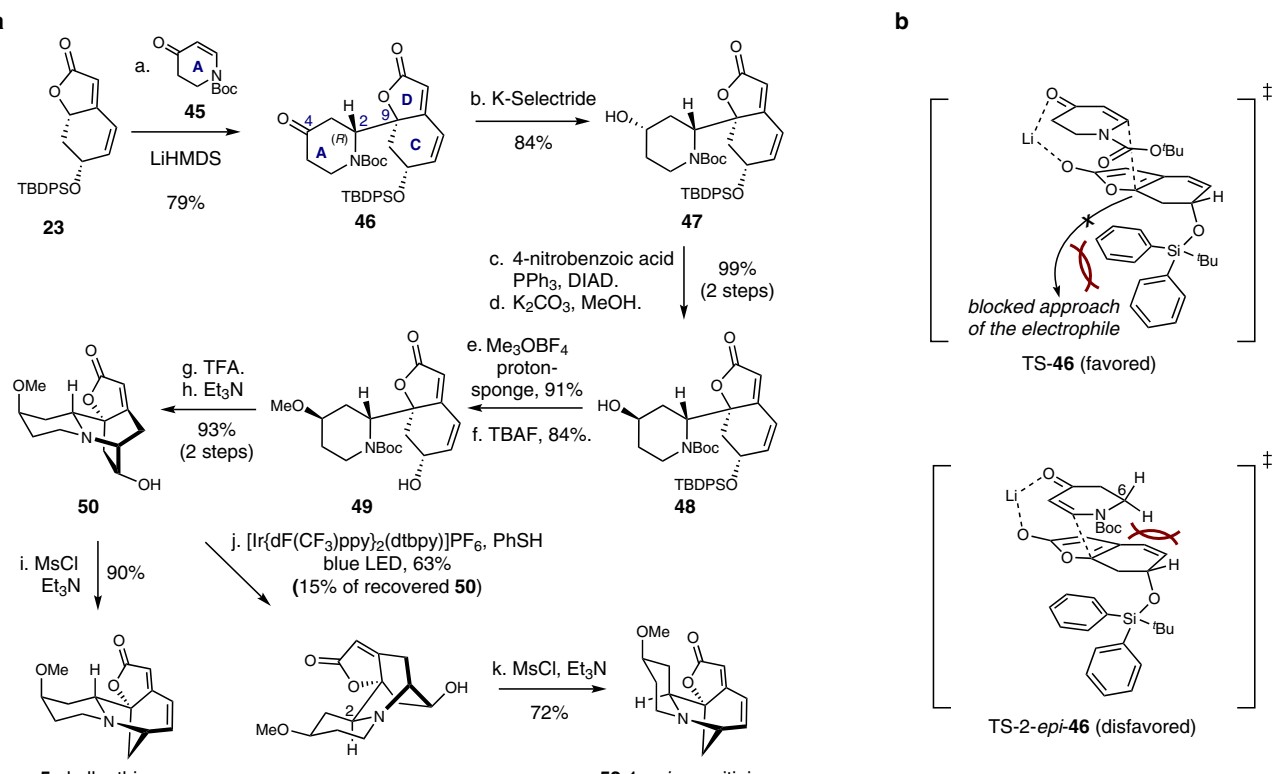

**Fig. 8 | Total synthesis of phyllanthine (5) and 4-epi-securitinine (52). a** Reagents and conditions: (a) **45** (1.5 equiv), LiHMDS (1.2 equiv), THF, −78 °C to −40 °C, 79%; (b) K-selectride (1.2 equiv), THF, −78 °C, 84%; (c) 4-nitrobenzoic acid (2.0 equiv), PPh₃ (1.5 equiv), DIAD (2.0 equiv), THF, 0 °C to 23 °C; (d) K₂CO₃ (3.0 equiv), MeOH, −78 °C, 99% (2 steps); (e) Me₃OBF₄ (3.0 equiv), proton-sponge (4.0 equiv), CH₂Cl₂, 0 °C to 23 °C, 91%; (f) TBAF (1.0 equiv), THF, 23 °C, 84%; (g) TFA:CH₂Cl₂ (1:1 *v/v*),

23 °C; (h) Et₃N:MeOH (1:2 *v/v*), 50 °C, 93% (2 steps); (i) MsCl (3.0 equiv), Et₃N (6.0 equiv), CH₂Cl₂, 0 °C, 90%; (j) [Ir{dF(CF₃)ppy}₂(dtbpy)]PF₆ (0.01 equiv), PhSH (1.0 equiv), blue LED (427 nm), MeOH, 23 °C, **51** 63% and **50** 15%; (k) MsCl (3.0 equiv), Et₃N (6.0 equiv), CH₂Cl₂, 0 °C, 72%; **b** Rationale for the diastereoselective formation of **46**. LiHMDS lithium bis(trimethylsilyl)amide, DIAD diisopropyl azodicarboxylate.

securinine-type alkaloids. Furthermore, our discoveries en route to monomeric high-oxidation state securinine-type alkaloids synthesis would serve as the basis for the exploration of even more complex high-order and high-oxidation state securinine-type alkaloids[6,8,11]. Those will be the subject of our forthcoming reports.

## Methods

### Experimental instrumentations

All reactions were performed in oven-dried or flame-dried round-bottomed flasks and vials. Unless otherwise noted, the flasks were fitted with rubber septa and reactions were conducted under a positive pressure of argon, and vials were tightly sealed with plastic septa and parafilm. Stainless steel syringes or cannula were used to transfer air- and moisture-sensitive liquids. Flash column chromatography was performed as described by Still et al. using silica gel (60-Å pore size, 40–63 μm, 4–6% $H_2O$ content, Merck). Analytical thin-layer chromatography (TLC) was performed using glass plates pre-coated with 0.25 mm silica gel impregnated with a fluorescent indicator (254 nm). Thin-layer chromatography plates were visualized by exposure to ultraviolet light, an aqueous ceric ammonium molybdate (CAM) solution, and/or an aqueous potassium permanganate ($KMnO_4$) solution.

### Reagents and solvents

Unless otherwise stated, all commercial reagents and solvents were used without additional purification with the following exceptions as indicated below. Dichloromethane and tetrahydrofuran were purchased from Merck and Daejung Inc., respectively and were purified by the method of Grubbs et al. under positive argon pressure.

### Data analysis

$^1H$ and $^{13}C$ nuclear magnetic resonance spectra were recorded with Bruker AVANCE NEO (500 MHz), Bruker AVANCE III HD Nanobay (400 MHz), Bruker AVANCE III HD (400 MHz), or Bruker AVANE NEO Nanobay (400 MHz) and calibrated by using the residual undeuterated chloroform ($\delta_H$ = 7.24 ppm) and $CDCl_3$ ($\delta_C$ = 77.23 ppm) or mono-deuterated dichloromethane ($\delta_H$ = 5.32 ppm) as internal references. Data are reported in the following manners: chemical shift in ppm [multiplicity (s = singlet, d = doublet, t = triplet, q = quartet, p = quintet, m = multiplet, app = apparent, br = broad), coupling constant(s) in Hertz, integration]. The NMR solvent $CDCl_3$ was taken from a stock containing anhydrous $K_2CO_3$ to remove residual DCl. High resolution mass spectra were obtained from KAIST Analysis Center for Research Advancement (Daejeon) by using ESI ionization method. Specific rotation was obtained by JASCO P-2000 polarimeter.

### Detailed experimental procedures

The detailed experimental procedures were provided in Supplementary Information.

### Cartesian coordinates and vibrational frequencies

Cartesian coordinates of the optimized geometries and vibrational frequencies of the optimized structures were provided in Supplementary Data 1.

## Data availability

Crystallographic data for the structure reported in this article have been deposited at the Cambridge Crystallographic Data Centre, under deposition number CCDC 2170237 (**9b**). Copies of the data can be obtained free of charge via https://www.ccdc.cam.ac.uk/structures/.

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

## Acknowledgements

This work was supported by the National Research Foundation of Korea (NRF) grant funded by the Korea Government (MSIT) (No. NRF-2021R1A2C2011203). We also acknowledge support by the National Research Foundation of Korea (NRF) grant funded by the Korea government (MSIT) (2018R1A5A1025208).

## Author contributions

S.H., S.P., and G.K. conceived the study. S.H. supervised the project. S.P. played a key role in experimentations. G.K. conducted computational studies. C.K. played a supportive role in experimentations. D.K. performed the single-crystal X-ray diffraction analysis. S.H., S.P., and G.K. wrote the paper.

## Competing interests

The authors declare no competing interests.
