## [Peer Review File · Nature Communications]

REVIEWER COMMENTS

Reviewer #1 (Remarks to the Author):

This article, submitted for publication in Nature Communication by Prof S. Han and co-workers, describes an important work around the total synthesis of several C4-oxygenated securinine-type alkaloids via a stereocontrolled diversification on the piperidine core.

The Han group has been one of the key players in the total synthesis of this natural substances family in recent years. They have notably already attained several major goals in the field such as, inter alia, the total synthesis of secu'amamine A & the fluvirosaones (Angew. Chem. Int. Ed. 2020, 59, 6894), the total synthesis of Flueggene D & I (Chem. Sci. 2020, 11, 10928), or even the total synthesis of the flueggiacotine B (J. Am. Chem. Soc. 2022, 144, 8932).

This new manuscript does not describe the synthesis of more complex structures compared to their previous reports, but still accounts a very important work towards yet-poorly described members. This lack of general and simple procedure to access such C4-oxygenated Securinega alkaloids is most certainly due to the extra-challenge of synthesizing compounds featuring one (or two in the case of the epoxidized members) stereogenic center on these complex tetracyclic molecules. Moreover, it should be emphasized that herein, the authors were not only targeting one specific natural substance, but aimed at an almost collective synthesis of this Securinega alkaloids sub-class.

In order to achieve these several goals, the simple access to key intermediates is mandatory. The authors have thus initially turned their attention to the asymmetric chemical production of either menisdaurilide and piperidine derivatives.

If the synthesis of menisdaurilide is already densely described in the literature, no really efficient enantioselective chemical synthesis of this compound has yet been reported. In fact, to date, the most efficient protocol for the production of this important intermediate has been reported by Peixoto and co-workers in 2019 in a racemic fashion, and through a 5 steps synthesis that can provide several grams with only one final purification. While these authors managed to separate both enantiomers using preparative HPLC techniques, the Han's group has recently been able to get enantiopure samples of menisdaurilide by resolution after forming chiral α -methoxyphenylacetic esters. This was notably made possible through several iterative purifications on silica gel chromatography. Considering the extra-efforts that such separations induce, a better solution was then desperately required in view of easily up-scaling the asymmetric chemical synthesis of these bioactive molecules. In these lines of thoughts, the authors report here a truly optimized way to access (-)-TBDPS-protected menisdaurilide by using a catalytic asymmetric desymmetrization of a meso-diol developed by Snapper and Hoveyda. A few steps are then required to produce highly

enantioenriched menisdaurilide on decagram scale, which represents an important improvement within the Securinega chemistry framework.

In view of producing C4-oxygenated Securinega alkaloids, it was also important to generate chiral piperidine moieties bearing a methoxy group or an epoxyde, prior to their coupling with menisdaurilide. This was done using either a Yun borylation (utilizing a Taniaphos complex) followed by a subsequent oxidation of the boronic ester generated, or a Shi epoxydation of piperidones.

With sufficient amounts of these different synthons in hand, the authors initiated their synthesis work. The first two targets were secu'amamine D and securigine A. These two compounds could then be obtained from their (neo)securinanes precursors by employing a Cope elimination or a 1,2-Meisenheimer rearrangement. Importantly, on this paragraph (and later in the article) the authors only cited the works of Magnus and Gademann for the skeletal rearrangement by a 1,2-nitrogen shift (neosecurinane=>securinane). In fact, Horii did the very first discovery of this rearrangement in 1965 [Chem. Pharm. Bull. (Tokyo) 1965, 13, 1307] during their viroallosecurinine synthesis campaign (MsCl, Pyridine then collidine). It should also be noted that ref. 30 corresponds to the only article that describes this rearrangement on all possible (neo)(nor)securinane diastereoisomers. Both these articles cannot remain under-cited on these aspects.

Once these two C4-methoxylated securinega alkaloids produced, the authors turned their attention to C3-C4 epoxidized members. Here another Mannich-type condensation was performed but using an epoxidized hemiaminal which leads to securigine C, confirming the revision of its configuration. This last natural substance was then oxidized using previously published conditions to furnish Securingine D.

The authors finally wished to target (-)-phyllantine, which displays a securinine-type skeleton featuring a R-configuration at C2. This was another difficulty because the Mannich condensation approaches used by Busqué, De March, and later by Gademann only generate S-configuration at C2 (high diastereoselectivity). In order to address this point, ref. 30 reported another approach relying on an "aldol way", where the C2 center was later formed by a reductive amination. However, surprisingly, when the piperidone was here reacted with the vinylogous lithium enolate derived from TBDPS-menisdaurilide, only the R-configuration on C2 was observed. In order to explain this differences in regard with classical Mannich condensation when using menisdaurilide enolates, a realistic mechanistic proposal is provided herein.

Last but not the least, along this manuscript, the authors also present a new strategy to address the question of the C2-center configuration by HAT-based stereogenic interconversions under photochemical conditions; a method recently developed by Ellman, Houk, and Mayer on simpler piperidine motifs. This method was found to be really useful, and could be performed on either securinane or neosecurinane members. Thanks to this tactic, 4-epi-phyllantine, and 4-epi-securitinine could then be finally obtained. It should still be noted that under such conditions, allosecurinine cannot be directly converted to securinine with high yields (allosecurinine is the

thermodynamic compound), and the production of the later thus still requires 2 extra-skeletal rearrangements from allosecurinine using photochemical processes.

To conclude, this reviewer is really impressed by the amount of work, the novelty, and the almost “collective” access to C4-oxidized Securinega alkaloids presented herein. This reviewer thus strongly recommends this manuscript for publication in Nature Communication after the revisions listed below:

Manuscript :

1- A better introduction for the precedents regarding menisdaurilide synthesis should be provided in order to explain the limitation of the different methods. The use of “in our hands” (line 72-73) is clearly inappropriate. It is mandatory to explain that decagram access to an enantiopure menisdaurilide-type intermediate still remain a challenge due to either enantiomers separation or lengthy asymmetric sequence.

2- The skeletal rearrangement neosecurinane => securinane has been first discovered by Horri and co-workers, and has been applied to all possible (neo)(nor)securinane diastereoisomers by the group of Peixoto and co-workers. Both these papers should be also cited when discussing these aspects.

3- Line 204-205 : The sentence about the control of the stereochemistry at C2 is too vague. A better explanation of the limitation of the Mannich strategy [i.e. too diastereoselective towards S-absolute configuration, and so almost exclusively leading to (–)-allosecurinine] is required. A discussion about other solutions found in literature, notably using an aldol strategy followed by a reductive amination (ref. 30), should also be specified.

4- The references need to be homogenized, and all authors must appear in each single citations (no “et al.”).

Supporting information: The data provided are of excellent quality. Only minimal corrections are required.

1- The authors do not mention how did they get the Hoveyda-Snapper catalyst used. If this catalyst was bought, the retailer should be specified. If they synthesized it, the reference used for its synthesis should be also added when mentioning it in the supporting information (compound 18 synthesis). A similar comment concerns the Taniaphos catalyst used for the synthesis of compound S3.

2- The description of the ¹³C spectrum of S1 contains the chloroform peaks. These extra peaks should be removed from the listing.

3- Many protocols mention a concentration "via air blowing". What occurs if this concentration is performed under reduced pressure at 40°C or at room temperature?

4- For readiness, a superimposition of the NMR spectra corresponding to the synthetic and literature-reported compounds could be added (not only tables).

Reviewer #2 (Remarks to the Author):

In the present manuscript, Han and co-workers reported a unified synthetic strategy which allowed them to collectively synthesize various C4-oxygenated securinine-type alkaloids. Several transformations reported inside are pretty interesting, i.e., an enantioselective desymmetrization reaction applied for their scalable preparation of TBDPS-protected menisdaurilide, as well as Ellman's HAT-based piperidine C2 epimerization for the interconversion between allosecurinine and securinine. However, I am still not very much in favor of the acceptance of this manuscript in its current form, mainly due to the following:

1. As the authors described in Figure 2, the present manuscript adopted heavily from several prior syntheses (ref. 19, 20, 21, 22) by incorporating with their own pre- and post-modification of the securinega framework. That is to say, the key vinylogous Mannich reaction using in-situ generated menisdaurilide derivative 11 and hemiaminals 28/37 is way too similar to previous reports by Busqué and de March (see ref. 19); The controlled 1,2-Meisenheimer rearrangement or Cope elimination highlighted in Figure 4 which generated secu'amamine D and securine A, respectively, again to a large extent, adopted from Magnus's seminal work (see ref. 20).

2. Relied upon a vinylogous annulation strategy (Mannich or Michael addition), the authors would be able to access seven C4-oxygenated securinine-type alkaloids. However, to achieve such goal, three de novo syntheses have also been performed, which partially impair the whole efficiency of this work.

Overall, the authors reported herein their extensive and unified synthesis of C4-oxygenated securinine-type alkaloids. However, owing to the concern of its insufficient novelty and efficiency, I am unable to recommend it for publication in Nature Communications.

General suggestions for SI:

1. If applicable, the authors should consider to include all the original NMR spectra of isolated and synthetic natural products, for the direct comparison.
2. The authors presented in their SI various NOESY spectra of key annulation intermediates, while detailed assignments are missing.

Reviewer #3 (Remarks to the Author):

The authors have contributed a clear, organized, professional and well written manuscript on the collective total synthesis of C4-oxygenated securinine-type alkaloids. The introduction is outstanding and describes the family of natural products, recent structural revisions of several of these members through the use of computation, and highlights key aspects of prior synthetic approaches to members of the family. Notably, only one synthesis of a securine alkaloid with C4-oxygenation has previously been reported. The authors then describe their carefully designed, concisely presented, and efficient syntheses to numerous C4-oxygenated securinine-type alkaloids. The syntheses incorporate a number of innovative and interesting features. For example, the authors developed a highly efficient asymmetric catalytic decagram scale route to 23 (Fig. 3), which they then applied to convergent and highly diastereoselective assemblies of key late-stage intermediates 29 (Fig. 4), 38 (Fig. 5) and 46 (Fig. 8). As another example, they performed impressive N-oxide 1,2-Meisenheimer rearrangements, Cope eliminations and Polonovski reactions as the final steps to several family members as depicted in Figs. 4 and 5. These syntheses are also noteworthy because they enabled the authors to employ total synthesis to achieve the first confirmations of the computationally proposed structural revisions for securinines C and D. The authors also utilized a recently reported photocatalytic hydrogen atom transfer (HAT)-mediated epimerization reaction to provide efficient

access to other alkaloid family members as well as epi-derivatives of other family members. The examples reported in this manuscript are by far the most complex molecules upon which this photocatalytic HAT epimerization has been performed. These examples therefore serve as important validating demonstrations of the method's applicability to architecturally complex natural product and/or drug structures.

In summary, the authors have contributed a professionally written manuscript that clearly describes a wealth of significant and innovative research. Consequently, this reviewer enthusiastically recommends publication without revision.

A. In response to reviewer 1

1. Reviewer's comment:

A better introduction for the precedents regarding menisdaurilide synthesis should be provided in order to explain the limitation of the different methods. The use of "in our hands" (line 72-73) is clearly inappropriate. It is mandatory to explain that decagram access to an enantiopure menisdaurilide-type intermediate still remain a challenge due to either enantiomers separation or lengthy asymmetric sequence.

Our revision:

Page 3, last paragraph

Before change:

Despite previous reports on enantioselective total synthesis of menisdaurilide,^{27,28,29,30} these routes were not amenable to large-scale synthesis in our hands.

After change:

Despite previous reports on enantioselective total synthesis of menisdaurilide,^{28,29,30,31} a practical decagram-scale synthetic process to enantiomerically enriched menisdaurilide or its derivative had not been described when we started our studies. In 2019, Peixoto and coworkers reported an elegant 5-step synthesis of (±)-O-^tbutyldimethylsilylmenisdaurilide that delivered 2.5 g of material in a single pass. However, enantiomerically enriched menisdaurilide derivative could be obtained only after semi-preparative chiral HPLC separation.³¹ Resolution of (±)-O-^tbutyldimethylsilylmenisdaurilide via its derivatization with the enantiomerically enriched carboxylic acid was possible but required multiple flash chromatographic separations.³²

2. Reviewer's comment:

The skeletal rearrangement neosecurinine => securinine has been first discovered by Horii and co-workers, and has been applied to all possible (neo)(nor)securinine diastereoisomers by the group of Peixoto and co-workers. Both these papers should be also cited when discussing these aspects.

Our revision:

Page 2, paragraph 1

Horii's references has been added.

Before change

Furthermore, Gademann's elegant total synthesis of allosecurinine (**2**), which was enabled by an intramolecular *aza*-Michael addition to access secu'amamine E (**13**) and its subsequent mesylation-based 1,2-amine shift (Fig. 2b)²⁰ has enabled us to explore new chemistries within the neosecurinine framework.²¹

After change

Furthermore, Gademann's elegant total synthesis of allosecurinine (**2**), which was enabled by an intramolecular *aza*-Michael addition to access secu'amamine E (**13**) and its subsequent mesylation-based 1,2-amine shift (Fig. 2b)^{20,21} has enabled us to explore new chemistries within the neosecurinane framework.²²

*Reference 20: Horii, Z., Ikeda, M., Tamura, Y., Saito, S., Kotera, K. & Iwamoto, T. Isolation of Securinol A, B, and C from *Securinega suffruticosa* Rehd. and the Structures of Securinol A and B. *Chem. Pharm. Bull.* **13**, 1307–1311 (1965).

Our revision:

Page 5, paragraph 2

Horii's and Peixoto's references have been added.

Before change:

On the other hand, when securitinine (**3**), accessed by treating **30** with MsCl and Et₃N,^{20,21}

After change:

On the other hand, when securitinine (**3**), accessed by treating **30** with MsCl and Et₃N,^{20,21,22,31}

Our revision:

Page 8, last sentence

Before change:

Subsequent Magnus²⁰ and Gademann-type²¹ 1,2-amine shift via a mesylation of the resulting hydroxyl moiety within compound **42** afforded the first synthetic sample of 4-*epi*-phyllanthine (**4**).

After change:

Subsequent 1,2-amine shift via a mesylation of the resulting hydroxyl moiety within compound **42** afforded the first synthetic sample of 4-*epi*-phyllanthine (**4**).

Our revision:

Page 6

Before change:

ent-Virosine B (**44**) was then transformed to securinine (**1**) upon mesylation of the alcohol moiety and consequent Magnus and Gademann-type 1,2-amine shift.

After change:

ent-Virosine B (**44**) was then transformed to securinine (**1**) upon mesylation of the alcohol moiety and consequent 1,2-amine shift.

3. Reviewer's comment

Line 204-205 : The sentence about the control of the stereochemistry at C2 is too vague. A better explanation of the limitation of the Mannich strategy [i.e. too diastereoselective towards *S*-absolute configuration, and so almost exclusively leading to (–)-allosecurinine] is required. A discussion about other solutions found in literature, notably using an aldol strategy followed by a reductive amination (ref. 30), should also be specified.

Our revision:

Page 10, paragraph 1

Before change:

The Mannich reaction-based union of the piperidine precursor (A ring) and the menisdaurilide derivative **23** has proven to be a robust method to set the *S*-configuration at the C2 position (Fig. 4 and 5).¹⁹ We also showed the versatility of HAT approach for the epimerization of this stereogenic center (Fig. 6 and 7).

After change:

The Mannich reaction-based union of the piperidine precursor (A ring) and the menisdaurilide derivative **23** has proven to be a robust method to set the *S*-configuration at the C2 position (Fig. 4 and 5).¹⁹ Peixoto and coworkers could obtain a diastereomeric mixture of compounds with *R*- and *S*-configurations at C2 via their aldol reaction-based fragments coupling and subsequent reductive amination strategy.³¹ We also showed the versatility of HAT approach for the epimerization of this stereogenic center (Fig. 6 and 7).

4. Reviewer's comment

The references need to be homogenized, and all authors must appear in each single citations (no "et al.").

Our response:

Below is the referencing style guide provided by Nature Communications.

<Nature Communications uses standard Nature referencing style. All authors should be included in reference lists unless there are six or more, in which case only the first author should be given, followed by 'et al. ' .>

We used "et al." for references that involve six or more authors. Hence, no changes have been made.

5. Reviewer's comment

The authors do not mention how did they get the Hoveyda-Snapper catalyst used. If this catalyst was bought, the retailer should be specified. If they synthesized it, the reference used for its synthesis should be also added when mentioning it in the supporting information (compound 18 synthesis). A similar comment concerns the Taniaphos catalyst used for the synthesis of compound S3.

Our revision:

Page S4

We added the following note at the end of page S4 of the supplementary information

Note: The synthesis of ent-16 was reported in <Nature 443, 67–70 (2006)>. Catalyst 16 was prepared using this protocol with enantiomeric starting materials.

Our revision:

Page S12

We added the following note at the end of page S12 of the supplementary information

Note: Taniaphos 25 was purchased from Strem Chemicals.

6. Reviewer's comment

The description of the ^{13}C spectrum of S1 contains the chloroform peaks. These extra peaks should be removed from the listing.

Our revision:

Page S5

Before change:

^{13}C NMR (126 MHz, CDCl_3): δ 164.6, 150.7, 136.4, 130.9, 124.3, 123.7, 123.5, 77.5, 77.2, 77.0, 74.1, 67.7, 32.8, 28.6, 25.9, 18.2, -4.4, -4.6.

After change:

^{13}C NMR (126 MHz, CDCl_3): δ 164.6, 150.7, 136.4, 130.9, 124.3, 123.7, 123.5, 74.1, 67.7, 32.8, 28.6, 25.9, 18.2, -4.4, -4.6.

7. Reviewer's comment

Many protocols mention a concentration "via air blowing". What occurs if this concentration is performed under reduced pressure at 40°C or at room temperature?

Our response:

Air blowing evaporation method was used for the removal of TFA/dichloromethane mixture after Boc deprotections. While, TFA and dichloromethane mixture could be also removed under reduced pressure using a rotovap, extensive washing of the instrument was required after its use because several co-workers in the group who are sharing the rotovap were conducting experiments using acids sensitive compounds. Hence, the TFA/dichloromethane mixture was removed via air blowing inside the fume hood.

8. Reviewer's comment

For readiness, a superimposition of the NMR spectra corresponding to the synthetic and literature-reported compounds could be added (not only tables).

Our revision:

The following section has been added in the supplementary information.

5. Comparison of NMR spectra of authentic and synthetic natural products (only for cases where NMR spectra are provided in the isolation report).

7b securigine A

Figure S7. ^1H NMR of securigine A in CDCl_3 from the isolation report.³

Figure S8. ^1H NMR of our synthetic securigine A in CDCl_3 .

Parameter	Value
Solvent	CDCl_3
Spectrometer Frequency	500.23
Nucleus	^1H

7b securigine A

7b securinine A

Figure S9. ^{13}C NMR of securinine A in CDCl_3 from the isolation report.³

Figure S10. ^{13}C NMR of our synthetic securinine A in CDCl_3 .

8b securinine C

Figure S11. ^1H NMR of securinine C in CDCl_3 from the isolation report.³

Figure S12. ^1H NMR of our synthetic securinine C in CDCl_3 .

8b securinine C

8b securinine C

Figure S13. ^{13}C NMR of securinine C in CDCl_3 from the isolation report.³

Figure S14. ^{13}C NMR of our synthetic securinine C in CDCl_3 .

9b securinine D

Figure S15. ^{13}C NMR of securinine D in CDCl_3 from the isolation report.³

Figure S16. ^{13}C NMR of our synthetic securinine D in CDCl_3 .

Parameter	Value
Solvent	CDCl_3
Spectrometer Frequency	400.12
Nucleus	^{13}C

9b securinine D

9b securigine D

Figure S17. ^{13}C NMR of securigine D in CDCl_3 from the isolation report.³

Figure S18. ^{13}C NMR of our synthetic securigine D in CDCl_3 .

B. In response to reviewer 2

1. Reviewer's comment:

If applicable, the authors should consider to include all the original NMR spectra of isolated and synthetic natural products, for the direct comparison.

Our revision:

See item 8 of reviewer 1.

2. Reviewer's comment:

The authors presented in their SI various NOESY spectra of key annulation intermediates, while detailed assignments are missing.

Our response:

Several compounds in our synthetic routes showed hindered rotation (especially those with the Boc protecting group) and appeared as a mixture of atropisomers. For those compounds, we verified that they actually are mixture of atropisomers by conducting ^1H NMR and NOESY experiment to obtain EXSY data. We added a note for compounds with EXSY data. See below for an exemplar note.

Note: ^1H -NMR spectrum shows two sets of signals, due to the presence of two rotamers in 80:20 ratio. This assignment was corroborated with the same ^1H -NMR and EXSY experiments, where exchange signals between absorptions of the same proton but corresponding to different rotamers, were observed. This behavior could also be observed in the ^{13}C -NMR spectrum.

REVIEWERS' COMMENTS

Reviewer #1 (Remarks to the Author):

This reviewer is fine with the corrections made, and wishes to congratulate once again the authors for the beautiful work reported herein.

[In comments to the Editorial office, Reviewer 1 affirmed that the authors had addressed the remaining concerns of the other referees.]